# Impact of Food Safety and Nutrition Knowledge on the Lifestyle of Young Poles—The Case of the Lublin Region

**Andrzej Soroka [1], Anna Katarzyna Mazurek-Kusiak [2,\*], Joanna Trafiałek [3], Agnieszka Godlewska [1], Joanna Hawlena [2], Surya Sasikumar Nair [3], Katarzyna Kachniarz [2] and Wojciech Kolanowski [4]**

[1]  Faculty of Medical and Health Sciences, Siedlce University of Natural Sciences and Humanities, Prusa Str. 14, 08-110 Siedlce, Poland; wachmistrz_soroka@o2.pl (A.S.); godlewskaa@uph.edu.pl (A.G.)

[2]  Department of Tourism and Recreation, University of Life Sciences in Lublin, Akademicka Str. 13, 20-950 Lublin, Poland; hawlena@interia.pl (J.H.); katkach@wp.pl (K.K.)

[3]  Department of Food Gastronomy and Food Hygiene, Institute of Human Nutrition Sciences, Warsaw University of Life Sciences—WULS, Nowoursynowska Str. 166, 02-787 Warsaw, Poland; joanna_trafialek@sggw.edu.pl (J.T.); surya_nair@sggw.edu.pl (S.S.N.)

[4]  Faculty of Health Sciences, Medical University of Lublin, Staszica Str. 4, 20-400 Lublin, Poland; wojciech.kolanowski@umlub.pl

\*  Correspondence: anna.mazurek@up.lublin.pl

**Abstract:** The aim of the study was to show the differences in healthy lifestyle and healthy food choices between high school and college students. The study was conducted in the Lublin region, Poland, with a group of 200 high school and college students using purposive sampling with the following four subgroups of 50 students, broken down by gender and type of school. Respondents completed a questionnaire concerning healthy lifestyle, healthy food choices, and barriers preventing a healthy lifestyle. Using discriminant analysis, the factors and barriers to practicing a healthy lifestyle and the factors of healthy food choices were identified by the respondent group and by gender. A multidimensional exploratory technique was also used to interpret the results. The surveyed high school and college students were not very committed to practicing a healthy lifestyle. Multidimensional exploratory technique was also used to interpret the lifestyle and healthy food choices questions. There was variation between the attitudes of college and high school students toward a healthy lifestyle. High school students paid more attention to physical activity and eating breakfast than did college students. On the other hand, college students, at a greater level than high school students, ate a healthy diet and checked the composition of the products they consumed, including the presence of preservatives and artificial additives, and the expiration date of the products. The main barriers to practicing a healthy lifestyle were, for college students, a lack of time and, for high school students, a lack of healthy food offerings in high school canteens.

**Keywords:** adolescents; healthy food choice; healthy lifestyle; students

## 1. Introduction

Science recognizes lifestyle as one of the critical factors influencing human health. However, the term healthy lifestyle has become a widely used slogan employed by various entities to stimulate the activity of social groups and direct them toward desirable behaviors [1,2]. Most people want to stay healthy and happy until later in life. Media coverage of irregularities in the food sector is causing a growing awareness among the Polish public of the health risks associated with food market choices [3]. Consumers expect food products to be fresh and organically produced, to have adequately selected ingredients, and at the same time to be free of preservatives and artificial additives [4].

The family environment, school environment, peers, mass media, and all medical services [5] significantly impact the formation of proper health attitudes in childhood. According to the WHO, as many as 75% of needs are met by the individual in the home

environment. Eating habits and healthy lifestyles are formed from childhood, which, along with the knowledge gained and skills acquired, influence the nature of health attitudes in adulthood [6,7].

However, often a young person does not follow the rules of a healthy lifestyle. It is in his nature to take risks. Young people need to experiment and test their abilities [8]. Adolescents like to show off in front of their peers and engage in risky behavior while negating everything adults say. This is all the more "cool" because the health consequences of risky eating behaviors are not revealed until many years later, and a person does not notice the connection, for example, between eating excessive amounts of unhealthy food in youth and a disease that "gets him" in adulthood [9]. The authors of [8] showed that adolescents are a group that is not very receptive to health education that would reduce the risk of many lifestyle diseases in adulthood.

A better understanding of the relationship between food choices, lifestyle, and adolescent health is important for developing behavioral changes and effective management programs to improve the health of young people [10].

At the end of the 20th century, programs promoting consumer responsibility for their health were initiated in Poland [11]. Poles became interested in the concept of healthy lifestyles. It led to a trend toward healthy lifestyles and healthy eating. Lifestyle is a complex category that, in some way, reflects the quality of life and informs how people live and spend their time and money. Lifestyle reflects cultural and subcultural patterns of behavior [12,13]. A healthy lifestyle often appears as a recommendation when improving health and quality of life [14]. However, it should also be noted that a lifestyle depends on several aspects that a person may not have the ability to affect. These are processes related to the features of the modern market economy, which include globalization and hyper-capitalism, growing awareness of the adverse and irreversible effects of modernization and digitization of societies, and the diminishing role of tradition in everyone's daily life [15]. Among health behaviors, one should note pro-health behaviors: physical activity, rational nutrition, taking care of personal hygiene, managing stress, proper interpersonal contact, reporting for preventive examinations, and avoiding anti-health behaviors, e.g., smoking, excessive consumption of alcohol, or use of psychogenic drugs [16]. Researchers have proven that preventive measures that motivate healthy lifestyles and safe eating have many health benefits in adulthood [17]. This trend is robust and continues to trend upward.

Obesity is one of the diseases of civilization and will become one of the most important nutritional diseases of our time [18]. In Poland, the problem of overweight and obesity concerns almost one-third of adolescents, and it is growing year by year. During 2014–2020, the percentage of adolescents with excessive body weight increased from 19.9% to 29% [19,20]. Of particular concern is the fact that obese adolescents are likely to remain obese well into adulthood and have an increased risk of developing diseases that will reduce the quality and length of their lives. Diet is one of the most important elements influencing human health and determining its functioning, in which quantity, quality, and method of food preparation are equally important [1,4]. Proper nutrition is essential in every period of life, but dietary irregularities, in particular, take a toll on the development of the young body. In general, a young, growing, and maturing body needs more building blocks than an adult. The body's energy and nutrient needs should be met with a balanced diet and healthy food choices [21].

The attention to a healthy diet and healthy food choices, as a prerequisite for health and long life, is increasing in Western societies [11]. It is estimated that approximately 30–50 disease entities are occurring in society, which is caused by unsatisfactory food quality and diet [22]. Excessive consumption of highly processed food, which has an improper balance of nutrients, contributes to an increase in the incidence of chronic diseases, such as diabetes, atherosclerosis, cardiovascular disease, obesity, and cancer [12,16]. It should be noted that the consequences of unhealthy eating are hazardous in adolescents during adolescence [14].

On the other hand, there is a flurry of advertisements in the media for unhealthy foods, e.g., sweets, chips, alcohol, and fast food. Young people have effortless access to unhealthy food and are "tempted" in various ways by producers and traders [23]. Polish eating habits often deviate from the ideal and global recommendations for proper nutrition [8]. Dietary errors among Polish adolescents are well known and widely disregarded, one of which is the failure among adolescents to eat breakfast. In Poland, about 23–30% of high school and college students skip their breakfast, and this group is dominated by girls. It is worth noting that, most often, the cause of disparities between the sexes is a misconception among young people that skipping breakfast is an effective method of weight loss [24]. Breakfast is the most important meal of the day; after a night's sleep, the blood glucose level is low, which impairs the function of the brain. This adversely affects cognitive processes, especially memory, and learning [25].

One possible reason for the low commitment to a healthy lifestyle among students is that it will be several years before making an immediate and significant effort, for example, giving up eating junk food, will yield visible effects, such as maintaining a healthy body weight and preventing chronic diseases in adulthood [9,22]. Many studies have confirmed that time perspective strongly influences starting or maintaining healthy lifestyles, such as physical activity, adhering to a healthy diet [26], or avoiding addictions [27].

The study hypothesis is that college students from the Lublin region are more aware of the impact of nutrition on human health and are more willing to make healthy food choices than high school students. Therefore, the study aims to show the differences in healthy lifestyle and healthy food choices between high school and college students.

## 2. Materials and Methods

The study was conducted in the Lublin region, Poland. The research was conducted for 12 months from January to December 2019 among a group of 200 male and female high school and college students, using purposive sampling with the following subgroups: 50 high school female students, 50 high school male students, 50 college male students, and 50 college female students (Table 1). A stratified sampling was used to reflect the representatives of Polish students' society in terms of gender; type of place of origin (rural areas, towns with up to 20,000 inhabitants, and cities with more than 20,000 inhabitants); and material situation (very good, good, average, bad, and very bad). These steps made it possible to determine the sample size, in which the confidence level was set at 0.95, and the maximum error was set at 0.05. Participant random selection was used, taking into account the availability of respondents. No other inclusion/exclusion criteria were applied. The questionnaires were collected until a certain number of correctly completed questionnaires were exhausted in the subgroups.

The bio-electric impedance analyzer Tanita SC-240 MA (Tanita Corporation, Tokyo, Japan) was used to analyze the participants' body fat content. The study conformed to the code of ethics of the World Medical Association and the standards for research recommendations of the Helsinki Declaration. Students agreed to participate in the study.

Respondents completed the questionnaire, which consisted of questions concerning lifestyle, healthy food choices, and barriers preventing a healthy lifestyle (Supplementary Materials). The questionnaire was specially designed for this study and was validated. Lifestyle, lifestyle barriers, and healthy food choices were assessed through a diagnostic survey method using a standardized face-to-face interview and 5-point scales, where 1—very low, 2—low, 3—moderate, 4—high, and 5—very high.

**Table 1.** Characteristics of the research sample (data in %, N = 200).

| Specification | | High School Students | | College Students | |
|---|---|---|---|---|---|
| | | Female | Male | Female | Male |
| Age | 17 years old | 26 | 24 | - | - |
| | 18 years old | 40 | 38 | - | - |
| | 19 years old | 36 | 38 | - | - |
| | 20 years old | - | - | 32 | 34 |
| | 21 years old | - | - | 34 | 34 |
| | 22 years old | - | - | 34 | 32 |
| Type of place of origin | rural areas | 36 | 38 | 36 | 38 |
| | Towns with up to 20,000 inhabitants | 34 | 32 | 34 | 32 |
| | Cities with more than 20,000 inhabitants | 30 | 30 | 30 | 30 |
| Material situation | Very bad | 10.00 | 10.00 | 10.00 | 10.00 |
| | Bad | 10.00 | 10.00 | 10.00 | 10.00 |
| | Average | 10.00 | 10.00 | 10.00 | 10.00 |
| | Good | 10.00 | 10.00 | 10.00 | 10.00 |
| | Very good | 10.00 | 10.00 | 10.00 | 10.00 |
| Total | % | 100 | 100 | 100 | 100 |

*Statistics*

Before starting the analyses, the normality of the distribution of each variable was examined. The variance matrices of the variables were homogeneous across the groups. Variations and relationships were considered statistically significant at $p < 0.05$.

The discriminant analysis was used in the study using Formula (1) [28].

$$D_{kj} = \beta_0 + \beta_1 x_{1kj} + \cdots + \beta_p x_{pkj} \tag{1}$$

where:

$n$—number of respondents,
$p$—number of variables,
$k$—number of cases,
$j$—number of groups,
$\beta$—function coefficient.

Using discriminant analysis, the factors and barriers to practicing a healthy lifestyle and the factors of purchasing healthy and safe food were identified by the respondent group and gender. Classification functions were also used in the study.

The study also used correspondence analysis to create a two-dimensional graph. (Formula (2)) [29]:

$$\Lambda = \sum_{i=1}^{r} \lambda^2 \tag{2}$$

where $r = \min(w,k) - 1$,

$w$—number of rows in a multidivisional table,
$k$—number of columns in a multidivisional table [26].

The study also used the chi-squared test and the V-Cramer coefficient. *Statistica* software, version 13 PL, was used for statistical calculations.

### 3. Results

The first stage of the study examined the relationship between the respondent group and body fat levels, BMI. Average body fat levels were present in 31.03% of female students, 26.44% of male students, 21.84% of female schoolchildren, and 20.69% of male schoolchildren. Body fat levels indicative of overweight were present in 30% of male students and 27.50% of female schoolchildren, and those indicative of obesity were present in 33.33% of female schoolchildren and 30.56% of male students. The respondent group significantly affected body fat levels ($p = 0.0003$, Chi$^2$ = 23.925). The C-Pearson contingency coefficient was 0.346, indicating a medium degree of correlation. Female students had significantly better body fat levels than female schoolchildren, and male students had better body fat levels than male schoolchildren (Table 2).

**Table 2.** Body fat levels of schoolchildren and students (data in %, N = 200).

| Body Fat Content * | High School Student | | College Students | | |
| --- | --- | --- | --- | --- | --- |
| | Female | Male | Female | Male | Total |
| Indicating underweight | 21.62 | 29.73 | 35.14 | 13.51 | 100 |
| Normal | 21.84 | 20.69 | 31.03 | 26.44 | 100 |
| Indicating overweight | 27.50 | 25.00 | 17.50 | 30.00 | 100 |
| Indicating obesity | 33.33 | 30.56 | 8.33 | 27.78 | 100 |
| Total | Chi-squared test = 23.925; $p = 0.0003$; C = 0.346 | | | | |

* When assessing the body fat content, the age and gender of the respondents were taken into account.

Normal BMI was more common among high school students than among college students (Table 3). BMI indicating overweight and obesity occurred among as many as 48.72% of male college students (vs. 20.51 females) and only among 10.26% of female high school students (vs. 20.51 males). Most of the female college students (55.56%) had a BMI indicating a thin body shape.

**Table 3.** BMI of schoolchildren and students (data in %, N = 200).

| BMI * | High School Student | | College Students | | |
| --- | --- | --- | --- | --- | --- |
| | Female | Male | Female | Male | Total |
| Thin | 22.22 | 22.22 | 55.56 | 0.00 | 100 |
| Normal | 28.95 | 26.32 | 24.34 | 20.39 | 100 |
| Overweight and obese | 10.26 | 20.51 | 20.51 | 48.72 | 100 |
| Total | Chi-squared test = 20.830; $p = 0.002$; C = 0.228 | | | | |

* When assessing the level of body fat, the age and gender of the respondents were taken into account.

In the second stage of the study, young respondents were asked to what extent they follow healthy lifestyle recommendations.

High school male students and college female students dominated the respondents with a high commitment to a healthy lifestyle. The young peoples' behavioral profiles were mapped mainly in the first dimension (72.35%). In contrast, college male students were characterized by a medium and low commitment to a healthy lifestyle (Figure 1).

The total inertia of the integral of the mass multiplied by the square of the distance from the center of gravity of the profiles indicated a relatively large dispersion. The profiles were mapped mainly in the first dimension (92.36%). One cluster was visible in the space. College female students as well as high school ones had a medium commitment to healthy food choices (Figure 2).

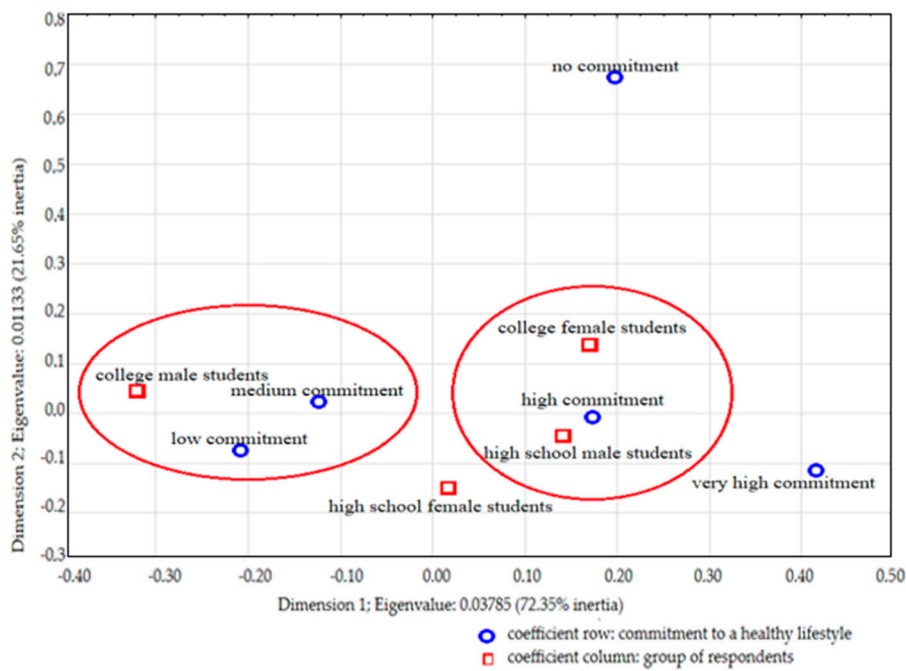

**Figure 1.** Evaluation of commitment to healthy lifestyles of high school and college students by gender.

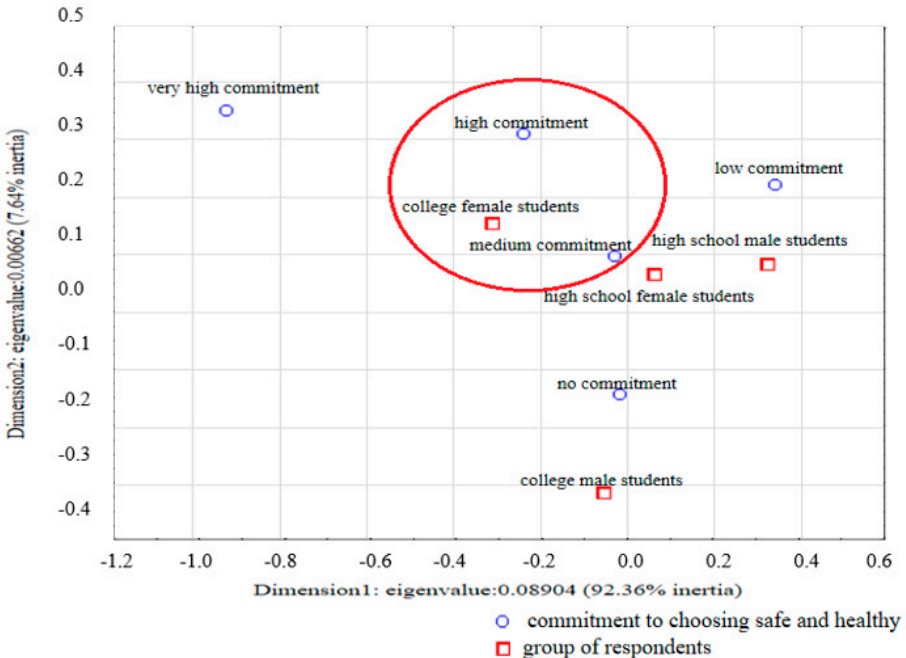

**Figure 2.** Evaluation of commitment to safe and healthy food choices of high school and college students by gender.

In the third part of the survey, students were asked to what extent they follow healthy lifestyle recommendations. Three of the six factors that were evaluated entered the discriminant function model.

The highest values of the classification function were reached with physical fitness. This factor was significantly more critical for high school students (1.905) than for college students (1.480). Eating breakfast every day was a significantly more important factor for high school students (1.071) than for college students (0.709). By far, the lowest values were observed in all the groups studied for the factor related to a healthy diet. This factor was

statistically more significant for college students (0.668) than for high school ones (0.560). College students were more aware of the impact of eating a healthy diet on their health status than were high school ones (Table 4).

**Table 4.** Differences in practicing a healthy lifestyle between high school students and college students.

| Factor | Discriminant Analysis Model: F(6.193) = 3.885; Wilks's λ: 0.892; *p* = 0.01 | | | | | | |
| | | | | Classification Functions | | Rating: Mean ± Standard Deviation | |
| | Wilks's λ | F | *p* | High School Students | College Students | High School Students | College Students |
|---|---|---|---|---|---|---|---|
| - eating a healthy diet | 0.621 | 6.178 | 0.014 * | 0.560 | 0.668 | 3.13 ± 1.09 | 3.17 ± 1.08 |
| - practicing physical fitness * | 0.637 | 9.761 | 0.002 * | 1.905 | 1.480 | 3.65 ± 1.10 | 3.15 ± 1.23 |
| - eating breakfast every day | 0.625 | 6.981 | 0.009 * | 1.071 | 0.709 | 3.72 ± 1.26 | 3.21 ± 1.32 |
| - don't using stimulants | 0.604 | 2.500 | 0.115 | 0.999 | 0.832 | 3.70 ± 1.54 | 3.30 ± 1.49 |
| - sleeping eight hours every day | 0.699 | 1.356 | 0.246 | 0.613 | 0.754 | 3.22 ± 1.36 | 3.19 ± 1.36 |
| - avoiding passive leisure time * | 0.693 | 0.131 | 0.718 | 1.300 | 1.250 | 2.89 ± 1.15 | 2.86 ± 1.19 |
| Constants | | | | −11.752 | −10.058 | | |

* Statistically significant differences ($p < 0.05$).

Considering the variation in a healthy lifestyle by gender of respondents, it can be seen that two of the six factors that were evaluated entered the discriminant function model. The highest values of the classification function were reached with the factor of physical fitness. This factor was significantly more important for adolescent male respondents (1.726) than for their female counterparts (1.277). Male respondents, both in college and high school, devoted most of their free time to practicing physical fitness. In contrast, it was significantly more critical for female respondents, both in college and high school, to avoid passive leisure time (1.446) (at the computer, TV, laptop) than for male respondents (1.109). Female respondents avoided passive leisure but spent less time than male respondents on physical activity. Wilks's λ for all these factors takes on high values (close to 1), so this variation was not very large between the genders of the subjects (Table 5).

**Table 5.** Differences in healthy lifestyle practices by gender of respondents.

| Factor | Discriminant Analysis Model: F(6.193) = 3.259; Wilks's λ: 0.907; *p* = 0.045 | | | | | | |
| | | | | Classification Functions | | Rating: Mean ± Standard Deviation | |
| | Wilks's λ | F | *p* | Females | Males | Females | Males |
|---|---|---|---|---|---|---|---|
| - eating a healthy diet | 0.818 | 2.043 | 0.155 | 1.053 | 0.817 | 3.26 ± 1.04 | 2.99 ± 1.13 |
| - practicing physical fitness * | 0.858 | 10.701 | <0.001 * | 1.277 | 1.726 | 3.24 ± 1.20 | 3.62 ± 1.15 |
| - eating breakfast every day | 0.916 | 1.609 | 0.206 | 0.850 | 0.675 | 3.57 ± 1.22 | 3.32 ± 1.14 |
| - don't using stimulants | 0.908 | 0.009 | 0.925 | 0.857 | 0.847 | 3.59 ± 1.54 | 3.37 ± 1.49 |
| - sleeping eight hours every day | 0.909 | 0.124 | 0.725 | 0.713 | 0.756 | 3.24 ± 1.38 | 3.16 ± 1.32 |
| - avoiding passive leisure time * | 0.835 | 5.834 | 0.017 * | 1.446 | 1.109 | 3.04 ± 1.14 | 2.63 ± 1.17 |
| Constants | | | | −10.729 | −10.436 | | |

* Statistically significant differences ($p < 0.05$).

In the next question, students were asked how they chose food products. Three of the six factors that were evaluated entered the discriminant function model. The classification function reached the highest values with checking the expiration date of products. This factor was significantly more critical for college students (1.579) than for high school ones (1.444). The model's highest discriminatory power (F = 14.081) and, at the same time, high values of the classification function were achieved with the factor checking the composition of food products and whether the products contain preservatives and artificial additives. This factor was also significantly more important for college students (1.509) than for high school ones (1.015). The lowest values of the classification function were observed with the knowledge of the food pyramid factor. This factor was statistically more significant for college students (1.439) than for high school ones (1.209). College students had more knowledge and better habits than high school ones in healthy food choices and were more aware of checking the expiration date and origin of food products (Table 6).

**Table 6.** Behaviors of high school students and college students in terms of healthy food choices.

| Factor | Discriminant Analysis Model: F(6.193) = 6.831; Wilks's λ: 0.824; *p* < 0.001 | | | | | | |
|---|---|---|---|---|---|---|---|
| | | | | Classification Functions | | Rating: Mean ± Standard Deviation | |
| | Wilks's λ | F | *p* | High School Students | College Students | High School Students | College Students |
| - checking the composition of food products | 0.785 | 14.081 | <0.001 | 1.015 | 1.509 | 2.09 ± 1.17 | 2.91 ± 1.25 |
| - paying attention to food certification | 0.726 | 0.178 | 0.674 | 0.307 | 0.369 | 1.93 ± 1.17 | 2.35 ± 1.19 |
| - checking the expiration date of food products * | 0.764 | 9.195 | 0.003 | 1.144 | 1.579 | 2.15 ± 1.19 | 2.81 ± 1.06 |
| - choosing organic food products | 0.832 | 1.699 | 0.194 | 1.078 | 0.874 | 1.96 ± 1.10 | 1.92 ± 1.04 |
| - checking the origin of food products | 0.725 | 0.099 | 0.753 | 1.020 | 1.080 | 1.57 ± 0.91 | 1.70 ± 0.87 |
| - following the food pyramid guidelines * | 0.743 | 4.199 | 0.042 | 1.209 | 1.439 | 2.51 ± 1.40 | 2.97 ± 0.87 |
| Constants | | | | −6.654 | −9.436 | | |

* Statistically significant differences (*p* < 0.05).

Statistically significant values of the classification function were observed for the factor of following the food pyramid guidelines and choosing organic food. These factors were statistically more significant for college female students (1.449 and 1.332) than for high school ones (1.090 and 1.002). Both college and high school female students had more knowledge and better habits in this regard than male students (Table 5). However, Wilks's λ for all of these factors takes values close to 1, so this variation was not very large between the genders of the subjects (Table 7).

**Table 7.** Behaviors of female and male high school and college students in terms of healthy food choices.

| Factor | Discriminant Analysis Model: F(6.193) = 2.817; Wilks's λ: 0.919; *p*-0.012 | | | | | | |
|---|---|---|---|---|---|---|---|
| | | | | Classification Functions | | Rating: Mean ± Standard Deviation | |
| | Wilks's λ | F | *p* | Females | Males | Females | Males |
| - checking the composition of food products | 0.819 | 0.002 | 0.968 | 0.979 | 0.974 | 2.51 ± 1.28 | 2.48 ± 1.28 |
| - paying attention to food certification | 0.822 | 0.440 | 0.508 | 0.370 | 0.276 | 2.22 ± 1.16 | 2.02 ± 1.25 |
| - checking the expiration date of food products | 0.825 | 1.105 | 0.294 | 1.004 | 1.150 | 2.46 ± 1.14 | 2.51 ± 1.23 |
| - choosing organic food products | 0.842 | 4.795 | 0.030 | 1.332 | 1.002 | 2.06 ± 1.13 | 1.75 ± 0.96 |
| - checking the origin of food products | 0.820 | 0.116 | 0.734 | 0.970 | 1.032 | 1.68 ± 0.86 | 1.57 ± 0.94 |
| - following the food pyramid guidelines * | 0.872 | 11.044 | 0.001 | 1.449 | 1.090 | 3.01 ± 1.39 | 2.35 ± 1.39 |
| Constants | | | | −7.776 | −6.795 | | |

* Statistically significant differences (*p* < 0.05).

In the last question of the survey, respondents were asked about barriers to practicing a healthy lifestyle. Two of the seven barriers that were assessed entered the discriminant function model. The analysis indicated that the barrier that most hindered the practice of a healthy lifestyle for college students was lack of time (9.082). For high school students, this barrier was the lack of healthy food offerings in school/college canteens (1.079) (Table 8). There were no significant differences in barriers to practicing a healthy lifestyle between college or high school female and male respondents (Table 9).

**Table 8.** Barriers to practicing a healthy lifestyle among high school students and college students.

| Barrier | Discriminant Analysis Model: F(7.192) = 2.373; Wilks's λ: 0.920; *p* = 0.024 | | | | | | |
| | Wilks's λ | F | *p* | Classification Functions | | Rating: Mean ± Standard Deviation | |
| | | | | High School Students | College Students | High School Students | College Students |
|---|---|---|---|---|---|---|---|
| - lack of time * | 0.741 | 4.262 | 0.040 | 8.663 | 9.082 | 4.38 ± 0.85 | 4.59 ± 0.67 |
| - lack of healthy food offerings in school/college canteens * | 0.656 | 7.458 | 0.007 | 1.079 | 0.622 | 2.39 ± 1,018 | 2.00 ± 0.99 |
| - lack of healthy lifestyle knowledge | 0.922 | 0.440 | 0.508 | 1.065 | 0.958 | 2.61 ± 1.04 | 2.48 ± 1.07 |
| - lack of motivation | 0.926 | 1.215 | 0.272 | 0.318 | 0.112 | 1.75 ± 1.02 | 1.60 ± 0.93 |
| - lack of food preparation skills | 0.920 | 0.023 | 0.881 | 1.377 | 1.397 | 2.41 ± 1.24 | 2.30 ± 1.19 |
| - I don't like healthy meals | 0.935 | 3.091 | 0.080 | 3.363 | 3.730 | 1.42 ± 0.751 | 1.46 ± 0.89 |
| - lack of money | 0.929 | 1.832 | 0.177 | −0.809 | −0.538 | 1.71 ± 0.96 | 1.78 ± 0.95 |
| Constants | | | | −25.978 | −27.288 | 4.47 ± 0.82 | 4.50 ± 0.69 |

* Statistically significant differences (*p* < 0.05).

**Table 9.** Barriers to practicing a healthy lifestyle by gender of respondents.

| Barrier | Discriminant Analysis Model: F(7.192) = 3.200; Wilks's λ: 0.895; *p* = 0.031 | | | | | | |
| | Wilks's λ | F | *p* | Classification Functions | | Rating: Mean ± Standard Deviation | |
| | | | | Females | Males | Females | Males |
|---|---|---|---|---|---|---|---|
| - lack of time * | 0.896 | 0.025 | 0.874 | 8.775 | 8.742 | 4.47 ± 0.82 | 4.50 ± 0.69 |
| - lack of healthy food offerings in school/college canteens * | 0.903 | 1.649 | 0.201 | 1.190 | 0.969 | 2.38 ± 1.08 | 1.92 ± 0.86 |
| - lack of healthy lifestyle knowledge | 0.897 | 0.323 | 0.571 | 1.130 | 1.035 | 2.70 ± 1.06 | 2.32 ± 1.02 |
| - lack of motivation | 0.907 | 2.416 | 0.122 | 0.550 | 0.250 | 1.88 ± 1.04 | 1.37 ± 0.79 |
| - lack of food preparation skills | 0.904 | 1.754 | 0.187 | 1.545 | 1.364 | 2.55 ± 1.26 | 2.07 ± 1.09 |
| - I don't like healthy meals | 0.897 | 0.369 | 0.544 | 3.553 | 3.422 | 1.55 ± 0.91 | 1.28 ± 0.65 |
| - lack of money | 0.898 | 0.571 | 0.451 | −0.615 | −0.771 | 1.92 ± 1.04 | 1.49 ± 0.78 |
| Constants | | | | −27.756 | −25.897 | | |

* Statistically significant differences (*p* < 0.05).

## 4. Discussion

According to Fijałkowska et al. (2019), the healthy lifestyle of Polish students is mainly determined by adequate nutrition, length of sleep, physical activity, and avoidance of risky behaviors [30]. As many as 44% use seasonal products, which are available at any store and are cheaper at the time. Nearly 70% of residents engage in physical activity, with 35% exercising at least once a week and 12% active daily. In contrast, Eurostat research [31] indicates that across the European Union, 26% of women and 36% of men engage in sports or other nonwork-related physical activity for at least 150 min a week. Adolescents are becoming less active with age due to the growing number of other, mainly sedentary, activities related to learning and leisure time. Moreover, global data consistently indicate inequalities in the physical activity level regarding age, gender, disability, socioeconomic status, or origin [32].

The presented results of the study on students from the Lublin region are in line with the contemporary trend of observing a decreasing interest in sports activity and are worrying in the context of the health of future generations. At the same time, the presented research shows that young respondents correctly associated physical activity with a healthy lifestyle. Other authors also obtained similar results [33–36]. Physical activity was an important factor for a healthy lifestyle for high school students compared to college students, and more important for male students than for female students. Such a result was consistent with the results of other authors studying students from sports and standard classes in a similar area of Poland [37]. These studies show that elementary school girls obtained weaker results in, for example, the run and jump test than boys. For girls and young women in the presented studies, it was important to avoid passive rest,

which is not the same as physical activity. This was most likely related to the fact that cleaning and cooking are more common among women than men [9]. Physical activity was more important for high school students than for college students, probably because high school students participated more systematically in physical education lessons than college students, and they have more free time for it.

Significant differences were noted in the eating of breakfast every day. High school students ate breakfast more often than college students. After all, usually, parents prepare breakfast for high school students and make sure they eat it. These results were consistent with those of Portuguese teenagers in many school years, who also did not skip breakfast [38], but contradictory to the results of Polish studies indicating skipping breakfast by teenagers, especially those not engaged in physical activity [39–41]. Other Polish research showed the same trend of skipping breakfast by adult Poles [42]. This study showed that high school students did not skip breakfast, though the tendency to skip breakfast increases as they get older. Such a trend differs from the prevailing trend, for example, in Spain, where the share of students who ate breakfast was very high and only slightly decreased during adolescence [43].

The surveyed college students, more than the high school ones, paid attention to selected aspects of healthy food choices. College students considered the composition of products without preservatives and artificial additives and the length of the expiration date. In this context, this research does not confirm the studies of Aronne et al. (2007) [44], Lobstein et al. (2015) [45], and Mazurek-Kusiak et al. (2019) [5], which showed that students are mainly guided by speed and simplicity when preparing meals and rarely consider the nutritional value of products, their freshness, and their composition. According to these authors, young people face many barriers to a healthy diet, both quantitatively and qualitatively, due to poor eating habits, the fast pace of life, and the lack of conditions and skills to prepare wholesome meals. However, studies by other Polish authors indicated that young adults paid attention to the quality of food products [46].

In contrast, the healthy food choices of Polish students are still unsatisfactory, which was also confirmed by studies conducted in other regions of Poland [47] and among American families with children [48,49]. This study did not show gender differences in analyzing food labels, as were shown in other studies [50]. College students showed more favorable healthy food choices than did high school ones. This was probably due to their age and higher level of education, as already indicated in the literature [51–54].

This study demonstrated that time and the availability of health-promoting foods were the main barriers to practicing a healthy lifestyle. Lack of time among young people as a barrier to a healthy lifestyle was demonstrated also by other authors [55,56]. College students have to divide their time between studying and working, and high school ones often cite the lack of healthy food on school grounds. School vending machines and convenience stores offered fast food and candy, and there was a lack of salads, fruits, and healthy snacks [57,58]. The availability of health-promoting foods for both high school and college students is very important, and the lack of such foods was often reported in the literature [59–62].

This research indicated that both high school and college students from the Lublin region showed many similarities with their peers in other parts of the world in terms of lifestyle trends. However, it was also shown which elements of their lifestyle might concern their future health.

However, this research has certain limitations. Firstly, it applies only to one area in Poland. High school and college students from other regions of Poland may have different knowledge about healthy food choices affecting their lifestyle. Secondly, high school and college students from other countries, especially developing countries, may differently perceive aspects of a healthy lifestyle and healthy food choices. Thirdly, the study did not consider the socioeconomic level of the families of the students surveyed, which might affect the findings.

## 5. Conclusions

The surveyed high school and college students were not very committed to practicing a healthy lifestyle and healthy food choices. Only a small number of respondents had normal body fat levels. There was variation between the attitudes of high school and college students toward a healthy lifestyle. High school students paid more attention to physical activity and eating breakfast than college students. On the other hand, college students more often than high school ones attempted a healthy diet and checked the composition of the products they consumed, including the presence of preservatives and artificial additives and the expiration date of the products. More efforts should be undertaken to foster more frequent physical activity and everyday breakfast consumption among high school and college students, as well as their healthy food choices. The most important barriers to practicing a healthy lifestyle were, for college students, a lack of time and, for high school ones, a lack of healthy food offerings in high school canteens.

## 6. Practical Implications

The research results can be generalized to other regions of Poland. Teachers in high schools and colleges should pass on knowledge about safe and healthy food. This should manifest in the curriculum and educational policy. Vending machines with sweets and salty snacks should be replaced in high schools and colleges with fruit and vegetable vending machines, and school shops should sell only healthy food. Local authorities should provide their residents free of charge with recreational sports facilities located in towns and villages. In addition, these facilities should have personal trainers and healthy eating advisors available to the local community.

**Supplementary Materials:** The following supporting information can be downloaded at: https://www.mdpi.com/article/10.3390/su151612132/s1.

**Author Contributions:** Conceptualization, A.K.M.-K. and J.T.; methodology, A.K.M.-K.; software, A.K.M.-K.; validation, J.T. and A.K.M.-K.; formal analysis, J.H.; investigation, A.K.M.-K. and A.S.; resources, A.G.; data curation, A.G. and A.S.; writing—original draft preparation, A.K.M.-K.; writing—review and editing, J.T. and W.K.; visualization, S.S.N.; supervision, A.K.M.-K., J.T. and W.K.; project administration, K.K.; funding acquisition, A.S. All authors have read and agreed to the published version of the manuscript.

**Funding:** This research received no external funding.

**Institutional Review Board Statement:** The study conforms to the code of ethics of the World Medical Association and the standards for research recommendations of the Helsinki Declaration. The protocol was approved by the local university ethics committee at the Siedlce University of Natural Sciences and Humanities no. 1/201.

**Informed Consent Statement:** Informed consent was obtained from all subjects involved in the study.

**Data Availability Statement:** Data are available upon request.

**Acknowledgments:** The authors would like to thank the participants for their consent to participate in the study.

**Conflicts of Interest:** The authors declare no conflict of interest.

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
