# Peer review of "Impact of Food Safety and Nutrition Knowledge on the Lifestyle of Young Poles—The Case of the Lublin Region"

_sustainability, doi:10.3390/su151612132_

Round 1
Reviewer 1 Report
Title: Impact of food safety and nutrition knowledge on the lifestyle of young Poles - the case of the Lublin Region
In the presented manuscript the authors aim to show the differences in healthy lifestyle and healthy food choices between high school and college students. With some limitations, the study is good designed. The work is interested and can be accepted for publication in the Sustainability but after major revision addressing the following points.
The hole text must be checked from English language editor, there a lot of grammatical errors.
Materials and Methods
- Is the Questionnaire original authors work or based on some previous researches?
- There is nothing mentioned about validation of used Questionnaire. Is it used without validation? Explain why exactly these 5 questions were chosen.
Discussion section
- Rows 294-295: Comparison of the studies among other regions of Poland and American families with children (study with very different design and study population) seems to be unsuitable here.
Conclusions
- Why do authors use the comment “Only a small part of respondents had normal body fat levels” in this section when nowhere previously assessment of body fat levels is mentioned as parameter that was concerned and measured in this study?
The hole text must be checked from English language editor, there a lot of grammatical errors.
Author Response
Thank you very much for taking the time to review and valuable comments. All comments from the reviewer have been incorporated into the revised article. Please see the attachment.

Reviewer 2 Report
Dear Authors,
The purposed manuscript entitled "Impact of food safety and nutrition knowledge on the lifestyle 2 of young Poles - the case of the Lublin Region" resulted interesting, and its scientific aims were well-explained.
However, I highly suggest a deep English revision performed by an English native speaker.
Best regards.
Dear English Service Board,
I highly suggest a deep English revision performed by an English native speaker.
Best regards.
Author Response

(The authors gave the same response as above.)

Reviewer 3 Report
The paper is interesting and addresses a very topical issue. It would have been nice to address it in more depth by better analyzing what resistance students of different age groups have in adopting a healthy lifestyle. Perhaps with a more detailed questionnaire that could assess different areas of lifestyle separately.
Some clarification requests for the authors:
1) How was the sample selected?
2) Was the recruitment of students random or were inclusion/exclusion criteria applied?
3) The age groups are very broad: within them might there not be different degrees of awareness and independence among the students?
4) Were students' personal parameters, such as weight, BMI, body composition, assessed? The conclusions refer to "normal levels of fat mass," line 321.
I found some typos and spelling errors in the discussion, particularly between line 256 and line 270.
Thank you!
Author Response

(The authors gave the same response as above.)

Reviewer 4 Report
Thank you for the opportunity to read this paper on the impact of food safety and nutrition knowledge on the lifestyle of young Poles in the Lublin Region. While the research objective and results are not entirely new, they provide a few points about their geographical setting. The authors have to put their findings into context. The study doesn't make a particularly significant concrete contribution, although it could. Below I will discuss my comments and suggestions, which hopefully can help the authors improve the study.
· The introduction lacks a strong case for how this information could alter or assist general practice or policy efforts.
· Give some prevalence, incident, or number of obese/ overweight general population, adults or youngsters, and add evidence and existing literature to understand the landscape of the population's general food choices or lifestyle.
· Page 2, last paragraph about breakfast, there is a contradictory statement about eating breakfast with existing literature; first, what percentage of the adult Polish population is missing breakfast or % of high school students are eating it? It is essential to understand the comparison. It seems like you are trying to compare apples and oranges.
· Study design is missing from the method section.
· What was the timeline for conducting the survey? The manuscript mentioned 2019, but it is unclear whether it was in 1-2 months or 6-12 months, so it better includes something like this "over 24 months (January 2016–December 2016)."
· In the Method section, Table 1 can be included as an Appendix because it s a study questionnaire.
· No statement related to ethics approval and how they consented to research and publication in the method section. It should be included in the text as well.
· Did you collect the respondent's primary demographic, age, social status, and resident area? These are the most critical factor in making choices for food.
· Except for the Table 1 questionnaire, what other variable did you collect from the participants? Provide details.
· Descriptive stats are missing from the result section.
· Include a statement in the result section about the initial enrolment of students.
· In the discussion section, the first lines reference the KukuÅ‚a Report; this report is a recommendation for the general population, not only high school or college students, so how do you relate it?
· Add a statement about the Generalisability of the findings.
· How results can be used for general practices and policies.
· What are the implications of the study?
Generally, it is a well-written manuscript. However, a few places might require further clarification, such as page 9, last para, which helps the reader to understand the author's perspective.
Author Response

(The authors gave the same response as above.)

Round 2
Reviewer 1 Report
The manuscript has been sufficiently improved and in this form is suitable for publication in Sustainability.
Reviewer 4 Report
The authors adequately addressed my concerns to follow up on my feedback from the first peer review round. However, even in response to comments, there was an error in comment 2 i.e. word material spelled as "Mareial." Therefore it is essential to review the manuscript for spelling and grammar.
See comment above.